# An Early Warning Protection Method for Electric Vehicle Charging Based on the Hybrid Neural Network Model

**Xiaoyu Zheng** [1], **Dexin Gao** [1,*], **Zhenyu Zhu** [1] **and Qing Yang** [2]

[1] College of Automation and Electronic Engineering, Qingdao University of Science & Technology, Qingdao 266061, China; 4020040050@mails.qust.edu.cn (X.Z.); 2020040030@mails.qust.edu.cn (Z.Z.)

[2] College of Information Science and Technology, Qingdao University of Science & Technology, Qingdao 266061, China; 03390@qust.edu.cn

[*] Correspondence: gaodexin@qust.edu.cn; Tel.: +86-138-6480-2293

**Abstract:** During the charging process of the electric vehicle (EV), a spontaneous combustion accident may occur due to overheating of the battery, causing personal danger and property damage. To address the charging safety of EVs, this paper proposes a new hybrid EV charging process early warning protection method by combining Convolutional Long-Short Term Memory (ConvLSTM), the sliding window method, and the residual analysis method. The method is fully trained by extracting the deep features of EV charging data through ConvLSTM, eliminating the influence of erroneous transmission data through the sliding window method, and setting a reasonable warning threshold through the residual analysis method. The cross-validation results showed that among the four training sets, the ConvLSTM model of training, set three, had the highest prediction accuracy compared with the CNN, LSTM, BiLSTM and CNN-LSTM models, with RMSE reaching 0.029, MAPE reaching 11.37, and $r^2$ reaching 0.89. Training set one had the worst prediction in the four training sets, and after using it to set the warning threshold, the alarm task was completed five sampling points earlier. Therefore, the hybrid model can quickly complete the safety warning task, thereby ensuring the safety of EV charging.

**Keywords:** electric vehicle; charging process; convolutional long-short term memory; safety early warning





## 1. Introduction

Compared with traditional vehicles, EVs have great advantages in energy conservation and emission reduction [1–3]. With the development of the industry, the safety of EVs has become one of the hot issues, which is not only the key to ensuring the safety of life and property, but also an important guarantee for the rapid development of the EV market. Realizing the safety early warning of the power battery of EVs and establishing a perfect mechanism have become the focus [4,5]. It has been found that the thermal runaway of the battery is an important cause of the spontaneous combustion of the EV charging [6,7]. Therefore, it is important to build an early warning model to protect the safety of EV charging.

At present, there are few research studies in the field of safety warnings for the entire EV charging process. Therefore, the early warning method of a single battery can be used for reference. Shah et al. [8] derived a dimensionless parameter thermal runaway number (TRN), whose value determines whether a lithium-ion battery will undergo the thermal runaway or not. This work laid the foundation for subsequent safety warning efforts. Lyu et al. [9] designed an online dynamic impedance measurement device for real-time overcharge warning and early thermal runaway prediction of lithium-ion batteries, which can effectively reduce the failure rate of thermal runaway. Jiang et al. [10] proposed a fault diagnosis and thermal runaway warning method of the lithium-ion battery pack with standard voltage as the identification object, which can achieve not only accurate

identification of faulty cells, but also the early detection of faults and early warning of thermal runaway.

Based on the research and analysis of thermal runaway for a single battery, a suitable early warning method was selected and applied to the entire battery of an EV to solve its combustion problem. As an emerging method in the field of machine learning, the neural network has successful practical cases in the fields of computer vision, early warning, and fault diagnosis. Therefore, the application of the neural network to early warning of the EV charging process has practical technical support. Since there are few applied cases of neural networks in the field of EV early warning at this stage, we can learn from other fields such as CNN [11–13], LSTM [14–16], BiLSTM [17–19], CNN-LSTM [20–22], and ConvLSTM [23–25] methods for research. In [11], a framework of automatically designed classifiers is proposed for the early detection of COVID-19 from chest X-ray images, minimizing redundant layers and improving prediction accuracy, which verifies the feasibility of CNN in the field of medication safety. In [14], based on the LSTM model, an early warning method for on-site earthquakes was proposed. Experiments show that this method can generate a highly nonlinear neural network and derive the alarm probability at each time step. This method can also effectively conduct an earthquake early warning, which verifies the reliability of LSTM in the field of geological safety. In [17], based on the BiLSTM model, a model for crop classification is proposed; it can fill in missing data and completes the classification task, which validates the accuracy of BiLSTM in the field of agricultural classification. In [20], a new data-driven method is proposed to use a hybrid deep neural network combining CNN, LSTM, and the classical neural network for RUL estimation, which verifies the validity of CNN-LSTM in the field of battery health status. In [23], the ConvLSTM model is used in traffic prediction, and the evaluation is based on the actual traffic data and the traffic flow data of the performance evaluation system, which verifies the applicability of ConvLSTM in the field of transportation convenience.

This research proposes a method for the EV charging safety warnings based on the ConvLSTM model. This hybrid neural network model uses the CNN model as the input of the LSTM model, which can solve the timing problem at the same time, train the model in more dimensions, and fully extract the deep features of the charging data [26,27]. It uses residual analysis to determine the EV charging status after obtaining the model residuals through the sliding window. Therefore, in theory, the model has the potential to predict charging faults. In the next part, the model is trained through the training set, and it is verified by comparing the predicted data with the true values in the test set. The main contributions can be summarized as follows:

(1) In terms of model structure; firstly, the Batch Normalization layer is added to speed up the convergence and prevent the gradient explosion; secondly, deep data features are extracted by overlaying two ConvLSTM cells; then the array is flattened with the Flatten layer; finally, a Dense layer is used to extract the correlation between the features after the nonlinear variation and map them to the output space.

(2) In terms of performance comparison, set the same parameters and compare the accuracy of the trained CNN, LSTM, BiLSTM, CNN-LSTM, and ConvLSTM models to verify the feasibility of the proposed ConvLSTM prediction model.

(3) In terms of practical applications, after the trained model meets the model accuracy, setting the thresholds can realize the model early warning and alarm tasks to predict the occurrence of faults and to effectively avoid charging accidents in EVs.

The rest of this research is organized as follows. Section 2 describes the EV charging system and communication information. Section 3 focuses on the model design and the specific process. Section 4 verifies the model feasibility through experimental analysis. We summarize the research in Section 5.

## 2. Problem Statement

### 2.1. Electric Vehicle Charging System Analysis

The EV charging platform system includes two parts: front end and backstage. The two parts cooperate to complete resource allocation and safety early warning work, which well guarantees the safety performance of the EV charging process. As shown in Figure 1, the direction of the arrow in the figure indicates the direction of data flow. The composition and functions of these two parts are as follows:

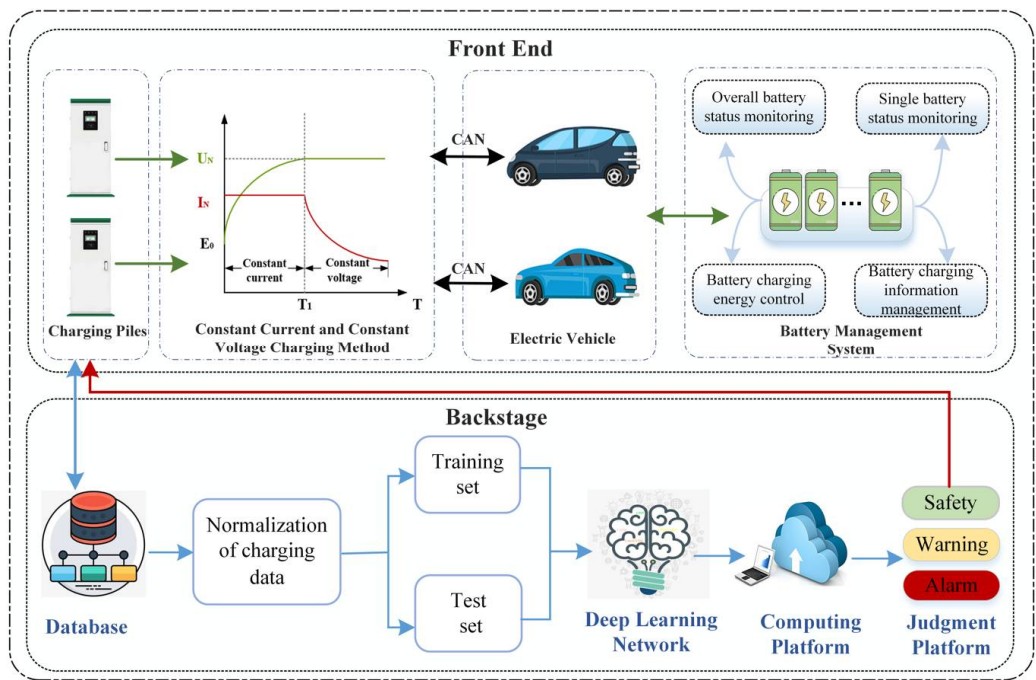

**Figure 1.** Diagram of charging platform system for electric vehicle.

Front End: This part is mainly composed of charging equipment and EVs. The charging pile uses the constant current and constant voltage stage charging method to supply power for different types of EVs [28]. It transmits the real-time charging data to the database and executes various control commands of the judgment platform. The Controller Area Network (CAN) communication protocol is adopted between the EV and the charging pile to ensure the real-time interaction of information. The battery management system provides real-time monitoring of single and overall battery status. When the battery temperature is abnormal, the judgment platform sends out corresponding commands to control the charging and stopping of the EV.

Backstage: This part is responsible for judging the charging operation state of the EV. The main charging data in the EV database is normalized and the data set is divided on this basis. The training set is used for the training of the ConvLSTM prediction model, and the test set is used for the judgment of the three states of the EV safety, early warning, and alarm.

### 2.2. Reference Basis for the Electric Vehicle Charging Process

In the global EV charging industry, there are many charging standards. Standards include interface standards, which relate to the fit of the connector, and current communication standards, which affect whether the plug can be energized when inserted. This section focuses on communication standards for EVs. The European Union mostly uses the European Norm (EN) standard; the charging standard is mainly IEC 61851-1 [29]. The United States uses the Society of Automotive Engineers (SAE) as the standard; the charging standard is mainly SAE J2293/2 [30]. In China's EV charging industry, the charging standard is mainly GB/T 27930 [31]. In the paper, the communication standard presented is

the Chinese standard GB/T 27930. As shown in Table 1, information such as the charging voltage, charging current and temperature of the EV can be obtained before the EV is charged. As shown in the battery management system part and the constant current and constant voltage charging method part in Figure 1, the main status information of the EV power battery before charging is sent to the charging pile through the CAN bus, and the charging pile will be adopted according to the rated current and rated voltage of the EV. The constant current and constant voltage charging method is used to charge the EV. From the starting point to the time *T*, the high-power constant current charging method is adopted, and the charging current is the rated charging current of the EV; at the time *T*, the rated voltage value is reached, so it is transferred to the next stage. At this time, the charging current gradually decreases. During the charging process, the EV sends the main status information of this phase to the charging pile in real-time and the information is shown in Table 2. If no abnormal condition occurs during the charging process, the EV battery pack will stop charging when it reaches 99.8% of the rated capacity to prevent the vehicle from burning due to overcharging.

**Table 1.** Main status information before charging of the electric vehicle battery.

| Start Byte | Word Length/B | Parameter | Unit | Precision |
|:---:|:---:|:---:|:---:|:---:|
| 1 | 2 | Vehicle power battery rated capacity | A·h | 0.1 |
| 3 | 2 | Vehicle power battery rated voltage | V | 0.1 |
| 5 | 2 | Vehicle power battery rated current | A | 0.1 |
| 7 | 2 | Vehicle power battery demand voltage | V | 0.1 |
| 9 | 2 | Vehicle power battery demand current | A | 0.1 |
| 11 | 2 | The maximum allowable voltage of the vehicle power battery | V | 0.1 |
| 13 | 2 | The maximum allowable current of the vehicle power battery | A | 0.1 |
| 15 | 2 | The maximum allowable temperature of the vehicle power battery | °C | 1 |
| 17 | 2 | Charging method: (the first stage: constant current charging; the second stage: constant voltage charging) | First stage: A Second stage:V | 0.1 |

**Table 2.** Main status information during charging of the electric vehicle battery.

| Start Byte | Word Length/B | Parameter | Unit | Precision |
|:---:|:---:|:---:|:---:|:---:|
| 1 | 2 | Charge voltage measurement | V | 0.1 |
| 3 | 2 | Charge current measurement | A | 0.1 |
| 5 | 2 | Charge temperature measurement | °C | 0.1 |
| 7 | 2 | Cumulative charging time | min | 1 |
| 9 | 2 | Estimate remaining charging time | min | 1 |

In Tables 1 and 2, the "Start byte" indicates the storage location for the data transfer process of the EV CAN communication protocol, and the "Word length/B" is the data storage range of the EV CAN communication protocol.

### 2.3. Analysis of the Model Selection Process

After reviewing the applications of neural network algorithms in different fields, the advantages and disadvantages of CNN, LSTM, BiLSTM, CNN-LSTM, and ConvLSTM were objectively evaluated, and the ConvLSTM model was finally chosen according to the actual EV charging application. In the actual working condition, the method proposed in this paper is different training models for different types of EV models, so the speed of model

training in the prenormal charging state of EVs should be considered when selecting the model. Since the ConvLSTM model can handle spatio-temporal features simultaneously and has a simple structure and fast training speed, the model is suitable for application to the field of real-time EV warning. After charging is completed, the work of the platform is to keep its model on file and update it in real time to facilitate rapid prediction and evaluation of the EV at that charging point next time. The comparison of the advantages and disadvantages of different models is shown in Table 3.

**Table 3.** Comparison of the advantages and disadvantages of different models.

| Methodology | Model Advantage | Model Disadvantage | Application Field | References |
|---|---|---|---|---|
| CNN | 1. Feature extraction can be automated; 2. Shared convolution kernel, can handle high-dimensional data. | 1. Training results do not easily converge to a global minimum; 2. Model improvement is more difficult due to encapsulation of feature extraction. | Charging safety / Fault Diagnosis | [12] / [13] |
| LSTM | 1. Long time memory function to solve sequence modeling problems; 2. Resolved the problem of gradient disappearance and gradient explosion. | 1. Disadvantages in parallel processing; 2. Average prediction compared to some of the latest networks. | Video Recognition / Economic forecasts | [15] / [16] |
| BiLSTM | 1. Information dependency can be captured in both directions; 2. More effective where two-way forecasting is required. | 1. Inability to transmit start-point information for overly long sequences well; 2. Inability to calculate the result of the next moment across the previous moment. | Power Dispatch / Wind speed forecast | [18] / [19] |
| CNN-LSTM | 1. Has the advantages of CNN and is widely used in feature engineering; 2. Has the advantages of LSTM and is widely used in time series. | 1. Unable to solve the prediction problem for bi-directional transmission; 2. Prediction effect limited by sequence length. | Battery Prognostics / Genetic Prediction | [21] / [22] |
| ConvLSTM | 1. Not only can temporal relationships be established, but also spatial features can be portrayed; 2. State-to-state switching can be converted into a convolutional calculation. | 1. Single time series problem, prediction results may not be as good as LSTM; 2. Single space series problem, prediction results may not be as good as CNN. | Video Detection / Fatigue Monitoring | [24] / [25] |

## 3. Design of Early Warning Hybrid Model for Charging Process

*3.1. Introduction to the Components of the Early Warning Hybrid Model*

The early warning protection method of the EV charging process is divided into four modules: charging data selection and processing, model building, sliding window, and state discrimination.

(1) **Data selection and preprocessing**: The charging data such as charging voltage, charging current, charging temperature, and charging time of the EV are transmitted to the charging pile through the CAN bus, and the charging pile transmits it to the backstage database. The backstage platform normalizes the charging data. The structure is shown in the data selection and preprocessing part of Figure 2.

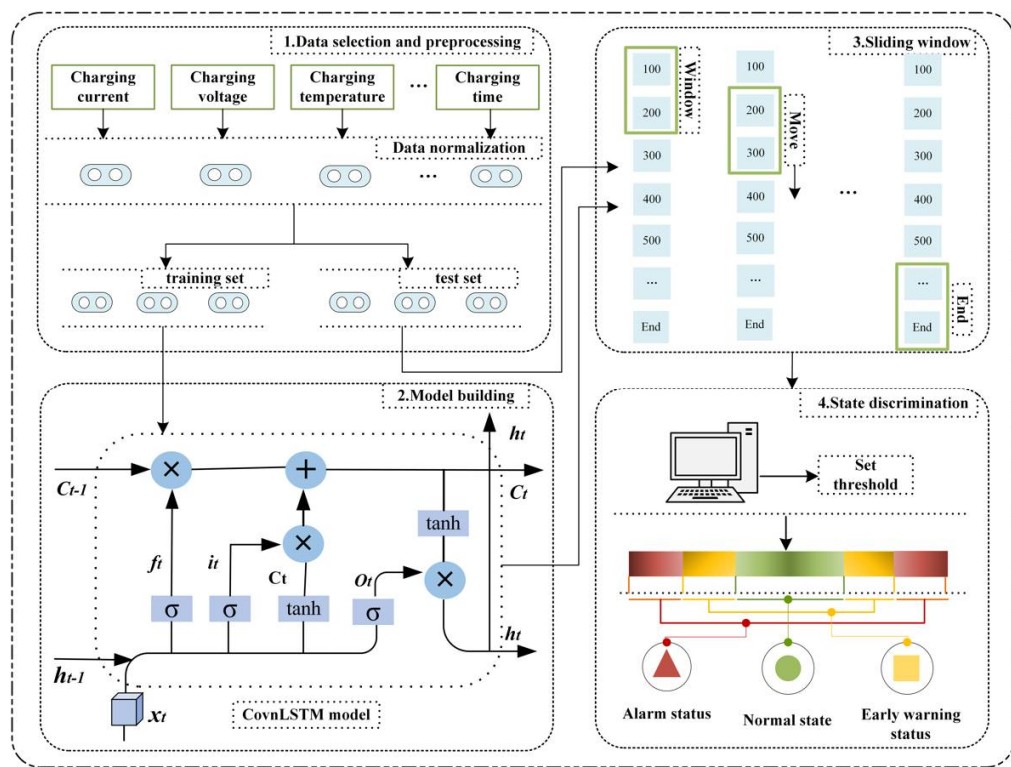

**Figure 2.** Diagram of safety pre-warning system for the electric vehicle charging.

Among them, the input features of the charging data are normalized and the data set is mapped between $[-1, 1]$ in order to prevent errors caused by data variation and to improve the accuracy of the ConvLSTM model. The calculation formula is:

$$X_{out} = \frac{x_i - x_{i\min}}{x_{i\max} - x_{i\min}} \tag{1}$$

where $x_i$ represents the actual value of the EV at the moment of charging $i$; $x_{i\max}$, $x_{i\min}$ represents the maximum and minimum values at time $i$ before normalization; $X_{out}$ represents $x_i$ normalized output charge value.

(2) **Model construction**: CNN is a neural network consisting of a convolutional layer, a pooling layer, and an output layer. It can share the weights of the convolutional kernel, reduce the free parameters, reduce the complexity of the network, and reduce overfitting, which has great advantages. It also has a powerful time series feature and extraction capability. Its calculation formula is:

$$c_t = f(W_{CNN} * n_t + b_{CNN}) \tag{2}$$

where $W_{CNN}$ represents the weight coefficient of the filter in the convolutional layer of the EV charging data; $n_t$ represents the EV charging data at time $t$; $*$ represents a convolution operation; $b_{CNN}$ represents the deviation coefficient of the convolution operation of EV charging data; $f$ represents the activation function of the EV convolution operation; $c_t$ represents the EV charging data sequence extracted after convolution.

The LSTM network consists of three main gate structures, i.e., the input gate, the output gate, and the forget gate. These three gates and activation function work together to filter information from historical data, retain useful information, and discard useless information. It has a strong learning ability. The structure is shown in the model building part of Figure 3. Its calculation formula is:

$$\begin{cases} i_t = \sigma(W_i \times [x_t, h_{t-1}, c_{t-1}] + b_i) \\ f_t = \sigma(W_f \times [x_t, h_{t-1}, c_{t-1}] + b_f) \\ c_t = f_t \times c_{t-1} + i_t \times tan\, h(W_c \times [x_t, h_{t-1}] + b_c) \\ o_t = \sigma(W_o \times [x_t, h_{t-1}, c_t] + b_o) \\ h_t = o_t \times tan\, h(c_t) \end{cases} \tag{3}$$

where $f_t$, $i_t$ and $o_t$ represent the forget gate, the input gate, and the output gate, respectively; $x_t$ represents the input data at $t$ time step; $c_t$ represents the status of the memory cell at $t$ time step; $h_t$ represents the output data at the previous time step; $\sigma$ represents the sigmoid function; $tan\, h$ is the activation function; $W$ represents the weight matrices; and $b$ represents the bias vectors.

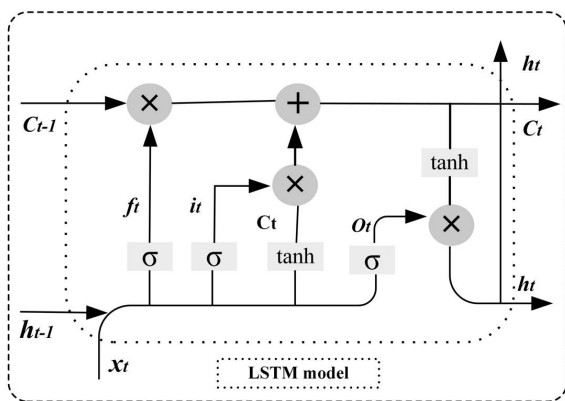

**Figure 3.** The structure of LSTM model.

The ConvLSTM uses CNN as part of the input of LSTM, different from the LSTM, the ConvLSTM uses convolutional operations, and the model is better able to extract deeper features of EV charging data. The structure is shown in the model building part of Figure 2. The updated formula is:

$$\begin{cases} i_t = \sigma(W_{xi} \odot x_t + W_{hi} \odot h_{t-1} + W_{ci} \circ c_{t-1} + b_i) \\ f_t = \sigma(W_{xf} \odot x_t + W_{hf} \odot h_{t-1} + W_{cf} \circ c_{t-1} + b_f) \\ c_t = f_t \circ c_{t-1} + i_t \circ tan\, h(W_{xc} \odot x_t + W_{hc} \odot h_{t-1} + b_c) \\ o_t = \sigma(W_{xo} \odot x_t + W_{ho} \odot h_{t-1} + W_{co} \circ c_t + b_o) \\ h_t = o_t \circ tan\, h(c_t) \end{cases} \tag{4}$$

where $\odot$ represents the convolution operations and $\circ$ represents the multiplication of the corresponding elements of the matrix, known as the Hadamard product.

(3) **Sliding window**: Using the sliding window analysis method, a sliding window of length $N$ is specified, and the data is processed through it. After the window slides forward one point, the predicted value is added to this window to generate a new window of the same sequence length. This process is repeated until the window is covered by the true value. Through the obtained several continuous sub-time series data, the prediction residual is continuously processed and analyzed to eliminate the influence of wrong charging data on the residual change during the transmission process, thereby avoiding false alarms effectively. The structure is shown in the Sliding window part of Figure 2.

When the width of the sliding window is $N$, the calculation formulas of the mean and standard deviation of the residuals under this window are:

$$\begin{cases} \overline{X} = \frac{1}{N} \sum\limits_{i=1}^{N} e_i \\ S = \sqrt{\frac{1}{N-1} \sum\limits_{i=1}^{N} (e_i - \overline{X})} \end{cases} \tag{5}$$

where $\overline{X}$ indicates the mean value of the residual of the charging temperature; $S$ is the standard deviation of the residual of the charging temperature; $e_i$ denotes the residual of the $i$th sampling point in the sliding window.

(4)  **Status discrimination**: The structure is shown in the status discrimination part of Figure 2. Using the sliding window to analyze and process the residual of the normal charging data and set the appropriate warning threshold. The formula for calculating the early warning threshold is:

$$\begin{cases} X_{E1} = k_1 |\overline{X}_{\max}| \\ X_{E2} = -k_1 |\overline{X}_{\max}| \\ S_E = k_2 S_{\max} \end{cases} \tag{6}$$

where $|\overline{X}_{\max}|$ is the maximum absolute value of the residual mean for charge temperature; $S_{\max}$ is the maximum value of the residual standard deviation for the charging temperature; $k_1$ represents the early warning coefficient of the residual mean; $k_2$ represents the early warning coefficient of the residual standard deviation; $X_{E1}$ and $X_{E2}$ represents the upper and lower early warning limits of the residual mean, respectively; $S_E$ is the upper early warning limit of the residual standard deviation.

The formula for calculating the alarm threshold is:

$$\begin{cases} X_{W1} = k_3 |\overline{X}_{\max}| \\ X_{W2} = -k_3 |\overline{X}_{\max}| \\ S_W = k_4 S_{\max} \end{cases} \tag{7}$$

where $k_3$ represents the alarm coefficient of the residual mean; $k_4$ represents the alarm coefficient of the residual standard deviation; $X_{W1}$ and $X_{W2}$ represents the upper and lower alarm limits of the residual mean, respectively; $S_W$ is the upper alarm limit of the residual standard deviation.

The specific EV charging status judgement criteria in this paper are shown in Table 4:

**Table 4.** Criteria for judging the charging status of electric vehicles.

| Electric Vehicle Charging Status | Discriminatory Criteria |
| :---: | :--- |
| Early warning status | (1)  $T_r > X_{E1}$ and $T_r > S_E$<br>(2)  $T_r < X_{E2}$ and $T_r > S_E$ |
| Alarm status | (1)  $T_r > X_{W1}$<br>(2)  $T_r < X_{W2}$<br>(3)  $T_r > S_W$ |

In Table 4, $T_r$ represents the temperature residuals, $X_{E1}$ represents the upper early warning limits of the residual mean, $X_{E2}$ represents the lower early warning limits of the residual mean, $S_E$ is the upper early warning limit of the residual standard deviation, $X_{W1}$ represents the upper alarm limits of the residual mean, $X_{W2}$ represents the lower alarm limits of the residual mean, $S_W$ is the upper alarm limit of the residual standard deviation. The criteria set in this paper are that when the temperature residual $T_r$ is above the upper limit of $S_E$ and at the same time above $X_{E1}$ or below $X_{E2}$, the state is judged to be an early warning state; and when $T_r$ exceeds any of the set alarm values, the state is judged to be an alarm state.

### 3.2. Introduction to the Early Warning Hybrid Model Process

The safety monitoring of EV charging systems needs to be realized by enhancing the dynamic monitoring and real-time early warning of EV charging equipment. The early warning hybrid model constructed by the ConvLSTM method can predict the potential risks during the battery charging process, so it can well guarantee the safety of the EV during the charging process. The security early warning process is mainly divided into

four stages: Data Processing, Model Training, Status Judgment, and Status Processing. The flow chart is shown in Figure 4.

**Figure 4.** Safe pre-warning flow chart for the electric vehicle charging.

The specific implementation process is as follows:

(a) **Data Processing**: Collect the front end EV charging data and transmit it to the back-stage database. Filter the charging data with reference significance, normalize and preprocess it, then divide the data set into a 25% training set and a 75% test set.

(b) **Model Training**: Use training set to determine LSTM and BiLSTM model parameters, train the corresponding model and output its evaluation standard values; determine CNN model parameters, train CNN, CNN-LSTM, and ConvLSTM models and output their evaluation standard values. If the model meets the model accuracy requirements, enter the next stage to set the EV temperature warning threshold; if not, return to retrain the corresponding model.

(c) **Status Judgment**: Set the EV temperature early warning and alarm thresholds. If its temperature is within the early warning threshold, this state indicates that the EV is charged normally and is the most ideal charging state. If its temperature is greater than the early warning threshold and less than the alarm threshold, the EV is in an early warning state at this time. If its temperature is greater than the alarm threshold, the EV is in an alarm state at this time.

(d) **Status Processing**: If the EV is in a normal charging state, no processing is required; if the EV is in an early warning charging state, the charging current will be reduced by 10% after an early warning signal is issued. If the EV is in an alarming state, the alarm signal will be issued, and then the charging power will be cut off to stop charging.

The whole process of judging can obtain the real-time capacity of the EV battery. If the real-time capacity is greater than 99.8% of the rated capacity, the charging will be stopped. The system is in high-speed closed-loop operation, which can complete the safety early warning task in time.

In Figure 4, $\overline{X}$ is the residual mean, $r^2$ is the decision factor, $C$ is the rated capacity, $T_1$ is the early warning thresholds, $T_2$ is the alarm thresholds, $S$ is the residual standard deviation, $C_1$ is the real-time capacity, RMSE is the Root-Mean-Square Error, MAPE is the

Mean Absolute Percentage Error, $T_r$ is charging temperature residuals, $k_1$ and $k_2$ are early warning threshold factors, $k_3$ and $k_4$ is the alarm threshold factor.

## 4. Experimental Verification and Analysis

### 4.1. Data Selection

This paper collects EV charging data through actual charging stations. The following is the introduction of the charging station: it adopts high-power IGBT type charging device, adopts IGBT super charging group technology, and has an output voltage range of 50~800 V, with high integration, high efficiency, low power consumption and high reliability, as shown in the Figure 5.

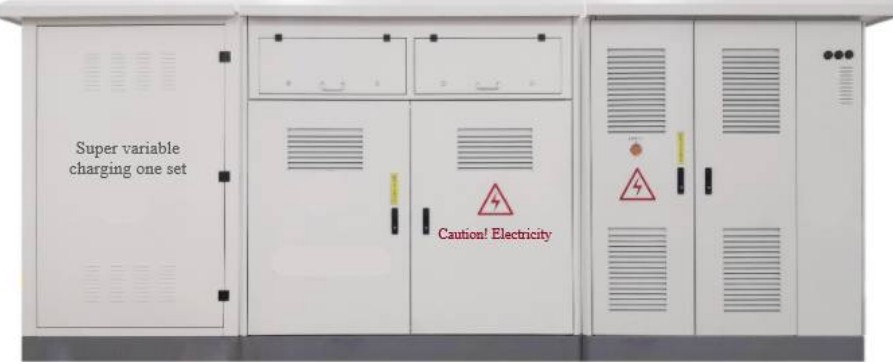

**Figure 5.** High-power IGBT type charging equipment.

In order to obtain reliable continuous charging data in the training model, after normalizing the actual charging data, the battery module in the Simulink library of Matlab R2020a software is used to simulate the battery model of the EV. The main parameters of the model are shown in Table 5:

**Table 5.** Parameters of simulated electric vehicle.

| Simulation Model Parameters | Numerical Values |
|---|---|
| Battery type | Lithium iron phosphate battery |
| Battery capacity/kA·h | 150 |
| Rated charging voltage/V | 412 |
| Rated charging current/A | 220 |
| Maximum allowable temperature/°C | 41 |
| Minimum allowable temperature/°C | −18 |

The EV adopts the high-power DC constant current and constant voltage charging method. On the charging side, based on the national standard communication standard GB/T 27930, the charging data of the interaction between the charging equipment and the EV BMS is dynamically collected and the transmission speed is 250 ms/sampling sites. After collecting the charging data, Simulink is used to simulate the model. Finally, the simulation data of 69,300 sampling sites are obtained.

To verify the effectiveness of the proposed method in predicting the charging temperature of EV batteries, the dataset was divided into a 25% training set and a 75% test set, and different parts of the dataset were used in turn to train the model in different iterations, with the training set selected as shown in Figure 6:

### 4.2. Model Construction and Evaluation Criteria

The simulation results for this part are performed in Python 3.8 and the Keras library. All model training runs on Windows 10 operating system, Intel® Core™ i5-1035G1 CPU @ 1.00 GHz 1.19 GHz (Manufacturer: Portland, OR, USA) processor. The platform version is

TensorFlow 2.4.1 and the program version is Python 3.8.1. The depth learning framework is Keras 2.4.3.

**Figure 6.** Diagram of training set selected.

The model in this paper is mainly composed of CNN and LSTM, and the prediction performance is further improved by using CNN as the input of LSTM. The model parameters involved are shown in Table 6:

**Table 6.** Specific parameters of ConvLSTM model.

| CNN Category | Parameters | LSTM Category | Parameters |
|---|---|---|---|
| Number of kernels | 32 | Cycle layers | 2 |
| Window size | 4 | Loop layer activation function | Tanh |
| Stride | 1 | Optimizer | Adam |
| Activation function | SELU | Neurons number | 90 |
| Pooling type | Global max pooling | | |

In Table 6, SELU is the activation function of the CNN model, and its main role is to map the input of the neuron to the output. Tanh is the activation function of the Loop layer of the LSTM model, and its role is to increase the nonlinearity of the neural network model. Adam acts as an optimizer for the LSTM, where a learning rate is maintained in each network circle and adaptively adjusted as the learning unfolds.

The internal structure of the model is shown in Figure 7, which consists of the ConvLSTM layer, Batch Normalization layer, Flatten layer, and Dense layer. Firstly, a Batch Normalization layer is added to speed up the convergence and prevent the gradient explosion. Secondly, deep data features are extracted by overlaying two ConvLSTM cells; then the array is flattened with the Flatten layer. Finally, a Dense layer is used to extract the correlation between the features after the nonlinear variation and map them to the output space.

The evaluation criteria in this research uses three common metrics for neural network regression prediction models: Root-Mean-Square Error (RMSE), Mean Absolute Percentage Error (MAPE), and $R$ Squared ($r^2$) to verify the accuracy of the prediction model. Its calculation formula is:

$$RMSE = \sqrt{\frac{1}{n} \sum_{i=1}^{n} (y_i - \hat{y}_i)^2} \tag{8}$$

$$MAPE = \frac{1}{n}\sum_{i=1}^{n}\left|\frac{\hat{y}_i - y_i}{\hat{y}_i}\right| \times 100 \tag{9}$$

$$r^2 = 1 - \frac{\sum_{i=1}^{n}(\hat{y}_i - y_i)^2}{\sum_{i=1}^{n}(\overline{y}_i - y_i)^2} \tag{10}$$

where $y_i$ represents the actual measured value of the charging temperature; $\hat{y}_i$ represents the predicted value corresponding to the charging temperature; $\overline{y}_i$ represents the average value of the output charging temperature. According to the above equation, the following conclusions can be drawn: The smaller the RMSE and MAPE while the larger the $r^2$, the better the model fitting effect.

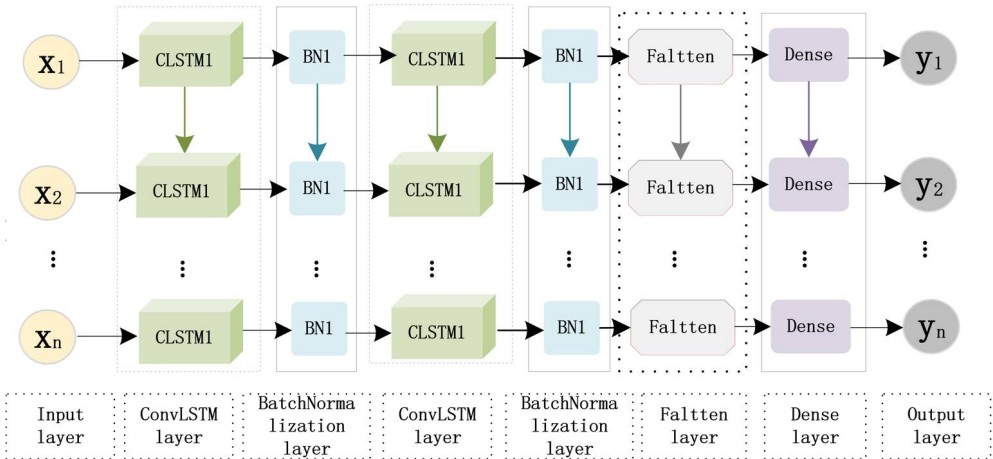

**Figure 7.** Diagram of the internal structure for the model.

### 4.3. Analysis of Prediction Experiment Results

The complete experimental process of this paper is as follows: Firstly, the actual charging data is normalized, and the Simulink module is used to simulate the EV battery to obtain continuous simulated charging data. Secondly, on this basis, the training set and the test set are divided. Then, the training set with the worst prediction result is selected for temperature residual analysis, and the corresponding early warning and alarm thresholds are calculated. Finally, the fault charging data is used to verify the EV charging warning method proposed in this paper. The following are the specific experimental process and results:

The normal charging data of the EV is selected to train the ConvLSTM model. When the EV is charged normally, the charging data is relatively stable, and the prediction error of the ConvLSTM model is relatively small. When a potential fault occurs in the charging of an EV, the degree of the fault will increase as the charging progresses, and the charging data of the EV will deviate from the normal charging range, resulting in a larger prediction error of the ConvLSTM model. To verify the accuracy and stability of the ConvLSTM model for predicting EV charging data, a cross-validation method was used after dividing the dataset. A graph of the predicted results of the cross-validation of this simulated EV model is shown in Figure 8:

As shown in Figure 8, by analyzing the graph of predicted results, the following conclusions are drawn:

(a) In the early stage of temperature prediction, the CNN-LSTM model in training set one gives predictions similar to the actual temperature values, and with more training data, the ConvLSTM model gives the best predictions in the middle and late stages of prediction.

(b)  In the early stage of temperature prediction, the ConvLSTM model in training set two was poorly predicted. The closest to the actual value was the LSTM model. However, in the middle and late stage, the ConvLSTM model and the BiLSTM model were the closest to the actual value.

(c)  The model prediction results of training set three are more satisfactory, and the ConvLSTM model proposed in this paper has the best prediction results in all three stages of prediction, which are closest to the actual values.

(d)  In the early and late stages of prediction, the ConvLSTM model and the BiLSTM model in training set four were closer to each other and had better predictions. However, the ConvLSTM model outperformed the BiLSTM model in the midterm prediction.

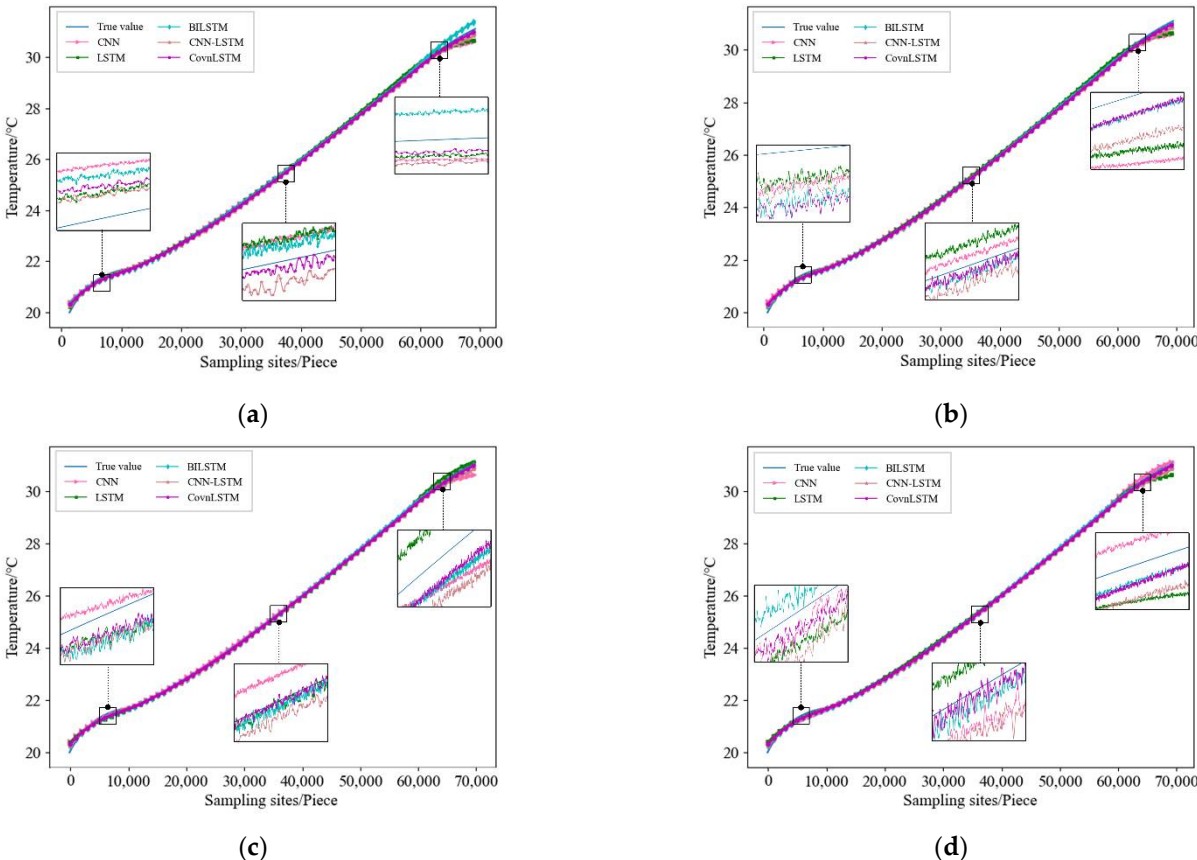

**Figure 8.** Diagram of cross-validated prediction results for simulation model. (**a**) Prediction result of training set 1. (**b**) Prediction result of training set 2. (**c**) Prediction result of training set 3. (**d**) Prediction result of training set 4.

The predictions from the cross-validation of this simulated EV model are shown in Table 7:

Table 7 summarizes the prediction errors of the simulation model and the experimental results are shown below:

(1)  Training set one: Using the top 25% of the data as training set one, the prediction accuracy of the different models was evaluated. The experimental results showed that the ConvLSTM models all outperformed the other four types of models in terms of prediction accuracy, with a 0.007 reduction in RMSE, a 1.66 reduction in MAPE, and a 0.18 improvement in $r^2$ compared to the CNN-LSTM models.

(2)  Training set two: The prediction accuracy of the different models was evaluated by training set two. The experimental results showed that the prediction accuracy of the models in training set two were all better than that in training set one. Regarding the ConvLSTM model in training set two, RMSE of was reduced by 0.003, MAPE was

reduced by 0.07, and $r^2$ was improved by 0.05 compared with that of the ConvLSTM in training set one.

(3) Training set three: Among the four training sets identified, training set three had the best prediction, with RMSE and MAPE reaching the lowest and $r^2$ the highest, where RMSE, MAPE, and $r^2$ were 0.029, 11.37 and 0.89, respectively.

(4) Training set four: In the four-part training set, training set four only predicted better than training set one. Compared with the ConvLSTM model in training set one, RMSE decreased by 0.01, MAPE decreased by 0.04, and $r^2$ improved by 0.04.

**Table 7.** Prediction results of cross-validation from simulation model.

| Data Sets | Algorithm Type | Evaluation Indicators | | |
|---|---|---|---|---|
| | | RMSE/°C | MAPE/% | $r^2$ |
| | CNN | 0.073 | 29.78 | 0.39 |
| | LSTM | 0.056 | 20.44 | 0.41 |
| Training set 1 | BiLSTM | 0.048 | 17.59 | 0.51 |
| | CNN-LSTM | 0.039 | 13.29 | 0.63 |
| | ConvLSTM | 0.032 | 11.63 | 0.81 |
| | CNN | 0.069 | 29.63 | 0.39 |
| | LSTM | 0.061 | 20.52 | 0.46 |
| Training set 2 | BiLSTM | 0.046 | 17.32 | 0.57 |
| | CNN-LSTM | 0.039 | 13.11 | 0.68 |
| | ConvLSTM | 0.029 | 11.56 | 0.86 |
| | CNN | 0.064 | 29.43 | 0.41 |
| | LSTM | 0.048 | 20.23 | 0.49 |
| Training set 3 | BiLSTM | 0.043 | 17.36 | 0.56 |
| | CNN-LSTM | 0.035 | 12.91 | 0.67 |
| | ConvLSTM | 0.029 | 11.37 | 0.89 |
| | CNN | 0.071 | 29.74 | 0.39 |
| | LSTM | 0.054 | 20.36 | 0.43 |
| Training set 4 | BiLSTM | 0.046 | 17.53 | 0.53 |
| | CNN-LSTM | 0.036 | 13.09 | 0.64 |
| | ConvLSTM | 0.031 | 11.59 | 0.85 |

The experimental results show that the ConvLSTM model has excellent performance in all four training sets using different evaluation metrics. RMSE, MAPE, and $r^2$ metrics outperformed the CNN, LSTM, BILSTM and CNN-LSTM network models. It is demonstrated that the ConvLSTM network model proposed in this paper improves the temperature prediction accuracy of the whole EV battery. Among them, training set three has the best prediction accuracy and training set one has the worst prediction accuracy. The following experiments all use the prediction results of training set one to validate the EV temperature warning model proposed in this paper.

The sliding window residual statistics method can continuously detect changes in the residual statistics characteristics in real-time. When abnormal conditions occur in temperature, its operating characteristics will change so that the new observation vector deviates from the normal operating state space; the charging status of EVs can be determined by calculating the warning and alarm thresholds of EVs [32–34]. The width $N$ was chosen to be 100. The mean value and the standard deviation of the temperature residuals for training set one of the simulation model are shown in Figure 9.

As shown in Figure 9a,b, the residual mean and the residual standard deviation can be obtained. According to many analyses of actual EV charging spontaneous combustion accidents, this paper takes $k_1$ and $k_2$ as 2, and calculates their early warning thresholds using Equation (6); when it exceeds the early warning threshold by 40%, an alarm signal is issued, this paper takes $k_3$ and $k_4$ as 2.8, and the alarm threshold is calculated using Equation (7). The specific values are shown in Table 8:

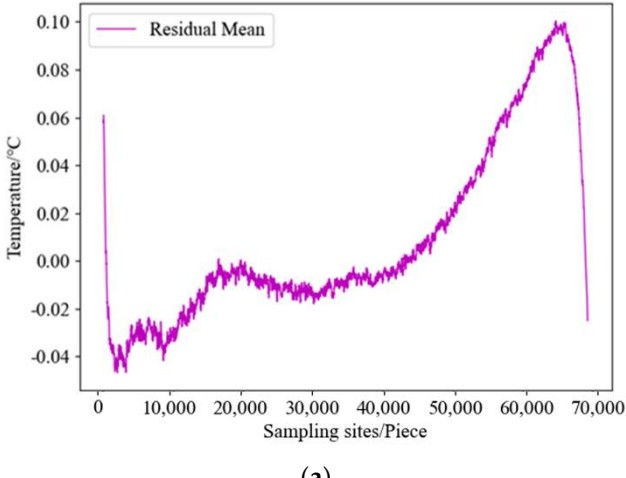
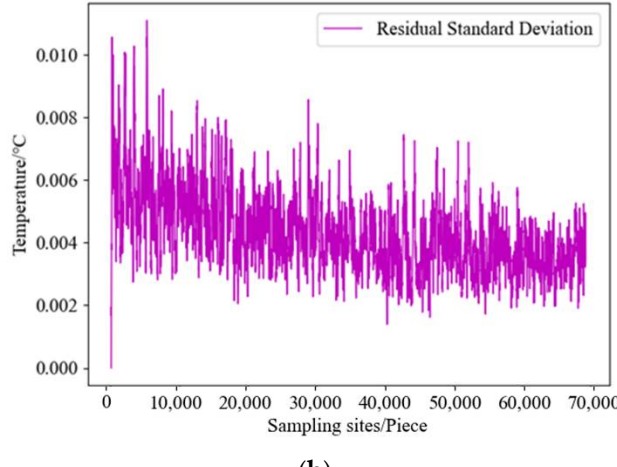

(**a**)                                       (**b**)

**Figure 9.** Residual results for training set 1 of the simulation model. (**a**) Residual mean. (**b**) Residual standard deviation.

**Table 8.** Calculated values for detailed pre-alarm thresholds.

| Residual Category | Parameters | Numerical Values/$^{\circ}$C |
|---|---|---|
| Residual mean ($\overline{X}$) Figure 9a | Minimum value ($\overline{X}_{min}$) | −0.0468 |
| | Maximum value ($\overline{X}_{max}$) | 0.1004 |
| | Maximum absolute value ($\lvert\overline{X}_{max}\rvert$) | 0.1004 |
| | The upper of early warning thresholds ($X_{E1}$) Equation (6) | 0.2008 |
| | The lower of early warning thresholds ($X_{E2}$) Equation (6) | −0.2008 |
| | The upper of alarm thresholds ($X_{w1}$) Equation (7) | 0.2811 |
| | The lower of alarm thresholds ($X_{w2}$) Equation (7) | −0.2811 |
| Residual standard deviation ($S$) Figure 9b | Maximum value ($S_{max}$) | 0.0111 |
| | Early warning thresholds ($S_E$) Equation (6) | 0.0222 |
| | Alarm thresholds ($S_W$) Equation (7) | 0.0311 |

Input the EV fault data into the ConvLSTM safety pre-alarm model, and the obtained charging safety pre-alarm results are shown in Figure 10.

As shown in Figure 10a, the residual mean exceeds the early warning threshold at the 54,250th sampling point and exceeds the alarm threshold at the 57,630th sampling point. As shown in Figure 10b, the residual standard deviation exceeds both the early warning and alarm threshold at the 39,998th sampling point. According to the safety pre-alarm method set in this research, the method sends an alarm signal at the 39,998th sampling point, and then stops charging. During the actual charging of the EV, the charging temperature at the 40,003rd sampling point was abnormal, and the on-board battery pack was burned at the 58,007th sampling point. Compared with the actual situation, the safety warning model proposed in this study stops charging 5 sampling points earlier than the earliest abnormal temperature point.

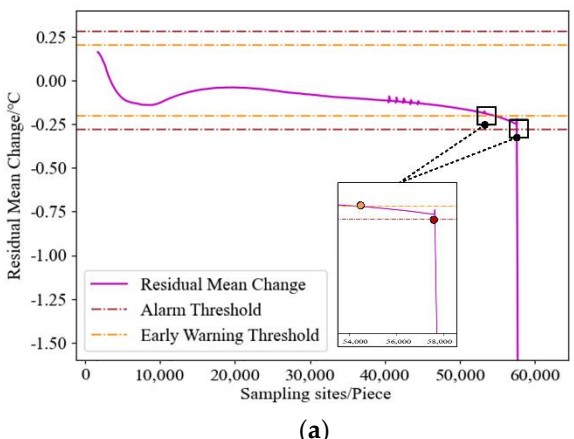
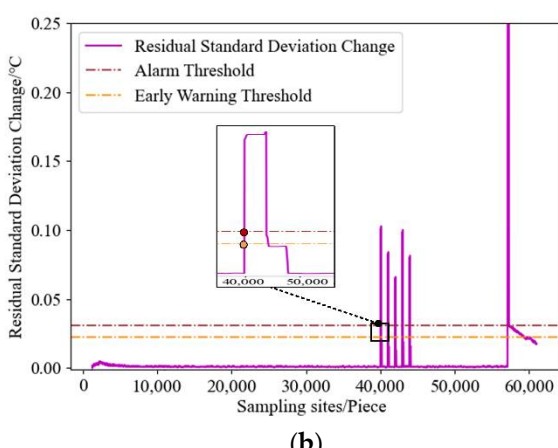

(**a**)   (**b**)

**Figure 10.** Safe pre-warning results for training set 1 of the simulation model. (**a**) Residual mean change. (**b**) Residual standard deviation change.

## 5. Conclusions

In this study, the ConvLSTM model is migrated to the field of EV warning for the problem of spontaneous combustion during EV charging, which is the first application of the model in this field.

Our experimental conclusions are as follows: The results showed that the model proposed in this study produced the lowest RMSE and MAPE while it obtained the highest $r^2$ compared to other models with the same parameters. Among the four training sets, the ConvLSTM model of training set three had the highest prediction, with RMSE reaching 0.029, MAPE reaching 11.37, and $r^2$ reaching 0.89. Training set one had the worst prediction and after using it to set the warning threshold, the simulation model completed the alarm task five sampling points ahead of schedule. Therefore, the early warning and alarm method proposed in this paper can detect abnormal charging conditions in advance and take corresponding protective measures such as reducing the current or stopping charging, effectively reducing the risk of spontaneous combustion in EVs.

In addition, the poorer prediction results for training set one is attributed to the larger charging voltage difference, higher charging current and more pronounced temperature variation at the early stage of EV charging; in training set four, it is because in the later stages of EV charging, the voltage difference kept narrowing, the charging current decreased and the temperature changes were not significant, training the model with these two parts of data reduced the prediction accuracy. Training set two and training set three were in the smoother charging stage, so the training model had better results.

In future work, the following two directions can be taken. Firstly, the early warning method proposed in this paper can be applied to the prediction of voltage, current and SOC of EVs, and the early warning indicators can collaborate with each other to make the EV early warning system more perfect. Secondly, the temperature prediction model can be replaced by a temperature rise prediction model, because the temperature rise is in the form of derivatives, which are more obvious and can capture abnormal charging conditions in a more timely manner. In summary, the method proposed in this paper has a more promising future in the field of EV early warning and hopefully will provide a new idea for researchers. There are, of course, shortcomings in our research. For example, our limited EV data allows only limited simulations to be performed to verify the accuracy of this hybrid model. In practical applications, consideration should also be given to data transmission delays, which are key to training the model in a timely manner, and this type of problem should be left to the collaboration of the relevant authorities.

**Author Contributions:** Conceptualization, X.Z. and D.G.; methodology, X.Z.; software, X.Z.; validation, X.Z.; formal analysis, X.Z., Q.Y. and Z.Z.; writing—original draft preparation, X.Z.; writing—

review and editing, Q.Y., Z.Z. and D.G.; visualization, X.Z. All authors have read and agreed to the published version of the manuscript.

**Funding:** This research was funded by the Key Research and Development Program of Shandong Province of China (Grant No. 2019GGX101012).

**Conflicts of Interest:** The authors declare no conflict of interest.

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
