# Peer review of "An Early Warning Protection Method for Electric Vehicle Charging Based on the Hybrid Neural Network Model"

_wevj, doi:10.3390/wevj13070128_

Round 1

Reviewer 1 Report

The work deals with very important issues related to the process of charging and supervision of electricity. The paper presents a very important model for warning and hazard situations resulting from the course of the charging process. In particular, a number of temperature factors resulting from the current and voltage parameters were taken into account. The introduction contains very interesting publications on electric vehicle charging systems. I would suggest supplementing the bibliography with the latest publications in this field. https://doi.org/10.3390/en14216859 https://doi.org/10.3390/en14123543; https://doi.org/10.3390/wevj13050073. Load standards analysis lines 107-111 - correct. Are there other standards? How does this relate to the model? Which one was used? The information in Chapter 3 is correct. The drawings are also well prepared. In particular, the model shown in Figure 4 is a great advantage of the work. In terms of formulas, I do not notice any errors. The assumptions are also correct. Chapter 4, well prepared, does not require any corrections. The summary is too extensive. I would recommend writing the most important conclusions in points. Give some numerical values ​​for the results achieved in chapter 4.

Author Response

Thank you for the comments concerning our manuscript entitled “An Early Warning Protection Method for Electric Vehicle Charging Based on Hybrid Model” (ID: wevj-1783307). These comments are all valuable and very helpful for revising and improving our paper, as well as the important guiding significance to our researchers. We have studied the comment carefully and have made correction which we hope meet with the approval. Since there are too many revisions, please check the attachment for details.

Reviewer 2 Report

Your article is oriented to area of charging electric vehicles and prediction of possible igniton. From the area of neuron network I have these remarks:

* your model was created from real measurement data, how many time, what number of different cars was used ?

* do you have some measurement with combustion effect ?

* what is or where is some impact from different outputs from your networks ?

From technical point of view I have several problems:

* LiFePo batteries critical temperature is over 41 degrees, am I right ?

* LiFePo batteries has BMS with checking bad temperature to stop charging process, how your model reacts ?

* LiIon is used very often in electric cars and they can burn, but is there same neuron model for different types of chemistry ?

Finally I don't see some new scientific results in your article. I think, if you can explain all questions above, we can find some relevant conclusion. In my eyes technically is your application quiet unusable.

Author Response

(The authors gave the same response as above.)

Reviewer 3 Report

Thank you for your efforts in this work. I would like to suggest some comments that I think might improve the presentation of this paper

1. Please do language editing/prof reading

2. The abstract can be improved to be more focus 

3. The literature review presented in very brief. Please do more critical review in this. 

4. In your revised critical review provide a table that compares critically what have been proposed so far. 

5. Section 2 : problem statment can be improved and extended. 

6. Add a comparison at the end of section 4 that critically compare your approach with others  

7. Rewrite the key findings of this research presented in the conlusion in points which will be more clear. Also, please mention what are the shortfall of your proposed method and what are the practical constraints 

Author Response

(The authors gave the same response as above.)

Round 2

Reviewer 2 Report

Dear authors, thank you for your new version and your comment. Finally I dont't agree with you in many fields but I think your work can open som wider discussion later.

* If you did not train model with negative scenario (fire) it can work in real (discussion result with my colleagues) because of missing BM behavior in this case.

* I still believe to Baterry management as the aprt of battery pack more then external computation.

* I am not able to find some big impact of used method for different training sets (small difference).

* I disagree critical temperature lies in EV. For me it is question of chemistry. I did small number of balancing modules and BMs.

* Prediction is well and for this reason I reccomend this article for publication.

Author Response

Thank you for recommending to publish our paper "An Early Warning Protection Method for Electric Vehicle Charging Based on Hybrid Neural Network Model" (ID: wevj-1783307). These comments are all valuable and very helpful for revising and improving our paper, as well as the important guiding significance to our researchers. We have studied the comment carefully and have made a correction which we hope meet with the approval.

Reviewer 3 Report

Thank you for your efforts by considering all the suggested comments. The quality of paper is much better now and from my side I am satisfied and don't have any further comments. 

Author Response

Thank you for your positive comments on our manuscript entitled “An Early Warning Protection Method for Electric Vehicle Charging Based on Hybrid Model” (ID: wevj-1783307).

First of all, thank you for your comments in the first round, which have important guiding significance for the revision and improvement of our paper; Secondly, thank you for recommending our paper for publication in the second round; Then, in our future work, we will continue to improve our EV early warning method with reference to your comments; Finally, I wish you good luck in your work and life.